# Mellow and Thick Taste of Pu−Erh Ripe Tea Based on Chemical Properties by Sensory−Directed Flavor Analysis

**DOI:** 10.3390/foods11152285

**Published:** 2022-07-31

**Authors:** Sihan Deng, Xinru Zhou, Haiyu Dong, Yongquan Xu, Ying Gao, Baijuan Wang, Xiaohui Liu

**Affiliations:** 1College of Tea Science, Yunnan Agriculture University, Kunming 650201, China; dengsihan9912@163.com (S.D.); zhouxinrue@163.com (X.Z.); dhy177356@163.com (H.D.); wangbaijuan123@126.com (B.W.); 2Tea Research Institute Chinese Academy of Agricultural Sciences, National Engineering Research Center for Tea Processing, Key Laboratory of Tea Biology and Resources Utilization, Ministry of Agriculture, 9 South Meiling Road, Hangzhou 310008, China; yqx33@126.com; 3Yunnan Organic Tea Industry Intelligent Engineering Research Center, Kunming 650201, China; 4Key Laboratory of Intelligent Organic Tea Garden Construction in Universities of Yunnan Province, Kunming 650201, China

**Keywords:** pu−erh ripe tea, mellow and thick, theabrownins, taste interaction, ultrafiltration

## Abstract

The mellow and thick taste is a unique characteristic of pu−erh ripe tea infusion, and it is closely related to the chemical composition of pu−erh ripe tea, which is less studied. This paper clarifies and compares the chemical composition of pu−erh ripe tea to that of the raw materials of sun−dried green tea, and uses membrane separation technology to separate pu−erh ripe tea into the rejection liquid and the filtration liquid. The results show that microorganisms transformed most physicochemical components, except caffeine, during the pile fermentation. It was found that total tea polyphenols, soluble proteins, total soluble sugars, theabrownin, and galloylated catechins became enriched in the rejection liquid, and the rejection liquid showed a more obvious mellow and thick characteristic. Taste interactions between crude protein, crude polysaccharide, and theabrownin were determined. They illustrated that the mellow and thick taste of pu−erh ripe tea with the addition of theabrownin increased from 4.45 to 5.13. It is of great significance to explore the chemical basis of the mellow and thick taste in pu−erh tea for guiding the pu−erh tea production process and for improving the quality of pu−erh tea.

## 1. Introduction

Pu−erh tea is a famous dark tea originating in Yunnan Province and is divided into pu−erh ripe tea (PRT) and pu−erh raw tea [1]. As a post−fermented tea, PRT undergoes pile−fermentation using Yunnan sun−dried green tea (SGT) as the raw material. The fermentation causes comprehensive chemical changes in the raw material, which contribute to the unique sensory quality and special health−care effects of PRT, such as anti−oxidation, anti−obesity, anti−hyperglycemia, and anti−hyperlipidemia [2,3,4]. In addition, microbial fermentation−mediated chemical transformation affects the taste quality of PRT. Mellow and thick taste is defined as a complex sensation of mellow, smooth, and sticky aftertaste. It benefits the consumer acceptability. However, the chemical mechanism is rarely studied.

The composition and content of chemicals of the tea infusion is the material basis determining the taste quality. At present, the understanding of the chemical composition of PRT has greatly advanced due to improvement in equipment. Polyphenols, oligosaccharides, and polysaccharides are identified as the main components that affect the taste quality of pu−erh tea [5]. Polyphenols in PRT are very different from those in unfermented tea. Part of the catechins undergo reactions such as oxidation, polymerization, and decomposition to form complex derivatives during the pile fermentation process, making PRT taste less bitter and astringent [6]. Some polyphenols in pu−erh tea are in the form of glycosides, in which polyphenols are linked to one or more sugar units through hydroxyl groups (O−glycosides) or carbon–carbon bonds (C−glycosides) [7]. Chen et al. [8] further confirmed that pu−erh tea polysaccharide conjugates were compounds formed by the combination of heteropolysaccharides (e.g., pectin) and a small number of short peptides. The UV absorption spectrum at 270 nm indicated that theabrownin might also be involved. The increase in aging time promotes the oxidation of polyphenols in pu−erh tea, and it enhances the conjugation between polysaccharides and protein. This phenomenon changes the taste of pu−erh tea from bitter to mellow [9]. So far, various compounds in PRT have been identified, and interactions between some components have been reported, but few studies have been conducted on the key contributors to the characteristic mellow and thick taste of PRT.

To investigate the formation mechanisms of the mellow and thick taste of PRT, the chemical compositions of the raw material (i.e., SGT) and corresponding final product (i.e., PRT) were compared in order to determine the differential chemicals. It is well known that pu−erh ripe tea contains various macromolecular polymers, such as tea polysaccharides, theabrownins, and pectin [10,11]. In this study, PRT infusion were separated into two (or three) phases by an ultrafiltration membrane, and the chemical materials of each fraction were identified. Then candidate components were added to the tea infusion to investigate the contributions. The findings will provide valuable information about the relationship between certain chemical materials and the mellow and thick taste. The findings will also provide support for the PRT process industry.

## 2. Material and Methods

### 2.1. Materials

SGT and PRT samples were provided by Yunnan Shuangjiang Mengku Tea Co., Ltd. (Lincang, China). Detailed information on the tea samples is supplied in Table 1. To ensure the accuracy of the experimental results, all the tea samples used came from the same batch. A microfiltration membrane (0.45 μm) was purchased from Shanghai Xinya Purification material factory (Shanghai, China). Ultrafiltration centrifuge tubes were purchased from Millipore (Billerica, MA, USA). Caffeine, gallic acid (GA), (−)−gallocatechin, (−)−epigallocatechin, (+)−catechin, (−)−epigallocatechin gallate, (−)−epicatechin, (−)−gallocatechin gallate, (−)−epicatechin gallate, (−)−catechin gallate, glutamic acid, Folin−Ciocalteu’s phenol, and acetonitrile were purchased from Sigma−Aldrich (Shanghai, China). N−Butanol and ethyl acetate were purchased from Shanghai Macklin Biochemical Co., Ltd. (Shanghai, China). Theabrownin (TB, 93%) was purchased from Yunnan Tangren Biotechnology Co., Ltd. (Honghe, China). Bradford Protein Assay Kit was purchased from Beyotime Biotechnology Co., Ltd. (Shanghai, China).

### 2.2. Preparation of Tea Infusions

Tea infusions were prepared according to the method introduced by Lin [12]. In brief, 3 g of tea leaves were mixed with 150 mL of boiling water, allowed to stand for 2 min, and then strained. The tea leaves were steeped with 150 mL of boiling water for a second time at once, allowed to stand for 5 min, and then strained. The two infusions were mixed to prepare the initial tea infusion. The PRT initial infusion was subsequently separated by ultrafiltering through a 10 kDa molecular weight cut−off (MWCO) membrane at 50 °C (5000 g, 20 min). The fraction that could not pass through the membrane was defined as the rejection liquid (RL), whereas the fraction that passed through the membrane was defined as the filtration liquid (FL). The RL was freeze−dried into powder for further tests.

### 2.3. Sensory Evaluation of Initial Tea Infusions

The initial tea infusions were cooled to 50 °C and evaluated by an expert panel. The infusions were scored by a trained team of five panelists. Sensory attributes were evaluated in terms of bitterness, astringency, mellow and thick taste, sweetness, and acceptability. Descriptive evaluation was defined by a ten−point scale: extremely strong (score range of 8–10), strong (score range of 6–8), neutral (score range of 4–6), weak (score range of 2–4), and extremely weak (score range of 0–2), and the acceptability was also scored 0–10.

### 2.4. Analysis of Chromatic Parameters (Color) of Initial Tea Infusions

The chromatic parameters of tea infusions were measured using a spectrophotometer (CM−3500d, Konica Minolta (China) Investment Ltd., Shanghai, China) introduced by the literature [13].

### 2.5. Determination of Total Tea Polyphenols (PP), Soluble Proteins (SP), Total Soluble Sugars (SS), Total Soluble Solids (TSS), and Theabrownin (TB)

The PP in tea samples were measured using the GB/T 8313−2018 method. The SP in tea samples were quantified using the Bradford Protein Assay Kit from Beyotime Institute of Biotechnology (Shanghai, China). The SS in tea samples were analyzed by the anthrone–sulfuric acid colorimetric method [14]. The TSS in tea samples were measured using a refractometer (RX−007α; Atago, Japan) after a zero calibration with water. The TB in tea samples was determined by Robert’s method [15], with modifications. Thirty milliliters of tea infusion was mixed with 30 mL of ethyl acetate in a 60 mL separating funnel and shaken for 5 min. Two milliliters of the aqueous layer was mixed with 2 mL of saturated oxalic acid and 6 mL of water, and then diluted to 25 mL with 95% ethanol. The absorbance at 380 nm was measured (Eb). The TB content was calculated using the following formula, where M is the moisture content:TB = 7.06 × 2Eb/(1 − M)

### 2.6. Analysis of Catechins, GA, and Caffeine

The catechins, GA, and caffeine in tea samples were determined by HPLC/UV (Shimadzu, Tokyo, Japan) as described by Cao’s method, with some modifications [13]. Tea samples were filtered through a 0.45 μm filter. The column Diamonsi^TM^ C18 (4.6 mm × 250 mm, 5 μm; Dikma Technologies Inc., Lake Forest, CA, USA) was used, and column temperature was set at 40 °C. The mobile phases were employed using 2% acetic acid in water (eluent A) and acetonitrile (eluent B). The linear gradient started with 6.5% B, 0–16 min; 15% B, 16–25 min; 6.5% B, 25–30 min. The flow rate was 1 mL/min, and detection wavelength was 280 nm.

### 2.7. Extraction of Crude Protein (CDB) and Crude Polysaccharide (CDT)

CDB was prepared following the procedure described by Tscheliessnig, with minor modifications [16]. The RL sample (10 mg) was mixed with 1 mL of NaCl (0.14 M), held for 6 hat 4 °C, then centrifuged at 4000× *g* for 15 min. Ice−precooled ethanol was added to the supernatants to reach a final concentration of 30.0% (*v/v*). The mixture was centrifuged at 4000× *g* for 15 min. The precipitates were dissolved in water to prepare 1 mg·mL^−1^ CDB solution.

CDT was prepared following the procedure described by Chen, with minor modifications [17]. The freeze−dried RL sample was dissolved in water to prepare a 10 mg·mL^−1^ solution, and then ethanol was added to reach a final ethanol concentration of 75.0% (*v/v*). The mixture was centrifuged at 4000 rpm for 15 min. The precipitates were configured to 1 mg·mL^−1^ CDT solution.

### 2.8. Taste Recombination Sensory Experiment

The effects of candidate components on the sensory quality of PRT infusion was designed as shown in Table 2, with PRT (P1) as reference. Sensory evaluation was conducted as described in Section 2.3.

### 2.9. Statistical Analysis

Three independent experiments were performed in triplicate and expressed as mean ± standard deviation. The analysis of significant difference between the means was carried out by one−way analysis of the variance (ANOVA), followed by the Duncan test to compare the means for significant variation with SPSS 23.0 (*p* < 0.05) (SPSS Inc., IBM Corporation, Armonk, NY, USA). The PLS−DA was conducted in Simca−P 13.0 software (Umetric, Umea, Sweden), and graphs were drawn by Origin 2021 software (OriginLab, Northampton, MA, USA).

## 3. Results and Discussion

### 3.1. Sensory Evaluation of SGT and PRT Initial Infusions

The results of sensory evaluation of SGT and PRT initial infusions are shown in Figure 1. The acceptability of PRT was higher than that of SGT (*p* < 0.05, Appendix A). Significant differences in the bitterness (*p* < 0.01), astringency (*p* < 0.01), mellow and thick taste (*p* < 0.01), and sweetness (*p* < 0.01) were found between the two infusions. Compared with SGT, the bitterness and astringency were attenuated, whereas the mellowness, thickness, and sweetness were enhanced in pu−erh ripe tea. It indicated that the pile fermentation led to significant changes in taste. The results were consistent with previous studies [10,18,19].

In addition to taste, the colors of SGT and PRT initial infusions (Figure 2A) were greatly different before and after pile fermentation. SGT was yellowish green, whereas PRT generally was reddish auburn. Chromatic parameters measured using a spectrophotometer (Figure 2B) were highly consistent with the results of visual observation. The CIE *L* a* b** system was used to describe the color of tea infusions, in which *L** represented lightness, and the absolute values of *a** and *b** represented the balance of red–green and yellow–blue tones, respectively [13]. The *L* a* b** values of SGT and PRT initial infusions were extremely different. The SGT initial infusion was brighter and greener, whereas the PRT initial infusion was redder. According to references, during the pile fermentation process, microorganisms and oxygen promoted the production of some high−molecular−weight compounds, such as TB. TB is reddish−brown and the most critical contributor to the color of PRT [10,20,21]. It is water−soluble and easily released from tea leaves to the infusion. Therefore, TB may also contribute to the characteristic color of PRT infusion.

### 3.2. Chemical Composition Analysis of SGT and PRT Initial Infusion

The contents of PP, SP, SS, TSS, TB, caffeine and catechins, which were reported to be important sensory components, are shown in Figure 3. A partial−least−squares discriminant analysis (PLS−DA) model based on the chemical compositions of SGT and PRT initial infusions was established. The results of four components:200 permutation tests (*R*^2^ = 0.279; *Q*^2^ = −0.121); *R*^2^*X* (cum), *R*^2^*Y* (cum), and *Q*^2^(cum) were 0.979, 0.998, and 0.996 (>0.5), suggested a good fit for the model. The clear discrimination of SGT and PRT initial infusions indicated that pile fermentation significantly changed the characteristic component. The TB content was significantly higher in the PRT initial infusion, whereas the contents of other measured components were significantly higher in the SGT initial infusion. The TB content increased significantly from 0.35% in SGT to 3.63% in PRT. This is consistent with a previous study [10]. In this study, the content of total galloylated catechins was reduced from 3.62% in SGT to 0.15% in PRT. EGCG, the most abundant galloylated catechin in SGT, was not detected in PRT. Galloylated catechins are considered to be the main contributors to bitterness and astringency, with GA contributing to the sweet aftertaste of green tea infusion following tannase hydrolysis [22]. The increase in TB content plays an important role in the formation of PRT quality [23]. As a result, the decrease in galloylated catechins and the increase in TB and GA might lead to reduced bitterness and astringency and enhanced mellow and thick taste of PRT.

It is reported that Maillard reactions, in which free amino acids react with soluble sugars, occur during the pile fermentation [10]. Therefore, it was unsurprising that a significant reduction in soluble sugars was observed in the PRT initial infusion as well, consistent with a previous report [14]. The soluble protein content also decreased. A possible reason was that proteins usually combined with tea polysaccharides through a covalent bond [8]. With the increase in tea polysaccharides after pile fermentation, soluble proteins were consumed, causing the decrease in soluble proteins [20].

### 3.3. Contributions of PRT, RL, and FL to the Mellow and Thick Taste

In Figure 4, the sensory evaluation of the PRT initial infusion, RL, and FL revealed that their tastes were very different (Appendix A). Among the attributes, the mellow and thick taste was most affected after the ultrafiltration treatment. It was greatly reduced in the FL and enhanced in the RL, indicating that RL might be critical for the mellow and thick taste. Compared with the PRT initial infusion, the RL was mellower and stickier, with a score of 6.68 over 4.45, and it had a caramel aroma and a pronounced stickiness. In contrast, the FL lost a large amount of flavor and the taste was plain and thin, but bitterness and astringency still existed.

### 3.4. Distribution of Chemical Components in RL and FL

The concentrations of PP, SP, SS, TSS, GA, caffeine, and catechins were determined and analyzed (Table 3). The PLS−DA model of each phase contained four components, as confirmed by 200 permutation tests (*R*^2^ = 0.229; *Q*^2^ = −0.439); *R*^2^*X* (cum), *R*^2^*Y* (cum), and *Q*^2^ (cum) were 0.961, 0.99, and 0.98 (>0.5) respectively, which indicated a good fit for the model (Figure 5). The results suggest that the chemical components were significantly affected by ultrafiltration. PP, SP, SS, and TB were mainly distributed in RL, reaching 1032.78 mg·L^−1^, 733.43 mg·L^−1^,1727.56 mg·L^−1^, and 5.62%, respectively. At the same time, the distribution in the FL was significantly reduced. Lu et al. [24] showed that amino acids such as proline or histidine in proteins could bind to polyphenol molecules through hydrophobic forces, and intermolecular hydrogen bonds could further enhance the binding. Among them, casein preferred to bind to highly polymerized polyphenols, whereas lactic protein preferred to bind to small molecular polyphenols. Xu et al. [25] suggested that the carbonyl groups of proteins might compete with other chemical components (e.g., caffeine) to bind to the hydroxyl groups of tea polyphenols. Proteins were mainly distributed in the RL and could bind to other substances to recruit them to the RL. It was confirmed by the finding of an increased total soluble solids in the RL. Further measurements showed that galloylated catechins were more distributed in the RL and less distributed in the FL. In contrast, non−galloylated catechins were less distributed in the RL, perhaps because they had fewer hydroxyl groups than galloylated catechins and lower affinity to proteins. The concentration of caffeine did not change significantly with a different infusion. Interestingly, the content of GA increased significantly in the comparison between SGT and PRT initial infusion. After ultrafiltration, the concentration of GA was not enriched in RL.

According to the results of the sensory evaluation of the tea infusion under ultrafiltration, the differences in bitterness and astringency scores among the three infusions were not significant. Previous studies [26] suggested that not only the concentrations of taste compounds and their interactions with other components affected the taste, but also their state in the system. According to studies reported on different food dispersion systems such as tuna (*Thunnus obesus*) head soup [27], white wine [28], and traditional Chinese medicinal herbs [29], the interactions of soluble polysaccharides with polyphenols or proteins in the dispersion system have an important influence on the sensory quality of food. Lei et al. [28] found that when polysaccharides, flavanols, and bovine serum proteins (BSA) coexisted, the presence of polysaccharides enhanced the polysaccharide–flavanol–BSA interaction, resulting in the alpha−helical structure of BSA becoming irregularly curled and even leading to protein precipitation, thus reducing the astringency of white wine. In this study, the RL was found to possess a more pronounced mellow and thick taste compared to the initial tea infusion. PP, SP, SS, and TB accumulated in the RL. TB is a polymer with great molecular weight and various structures, which are formed by oxidation and condensation of catechins and their catechin oxidation [30]. Table 3 shows that the content of TB reduced from 5.62% in RL to 1.12% in FL. Therefore, we believe that the RL also contained a large number of polymeric substances (TB) that cannot be separated through a 10 kDa membrane to molecular weight cut−off. In summary, we hypothesized that the mellow and thick taste of PRT might be related to complex particles formed by the interactions between proteins, polysaccharides, and TB in the tea infusion.

### 3.5. Taste Recombination Sensory Experiment

To test our hypothesis, TB and crude extracts of proteins and polysaccharides enriched in the RL were obtained and added to the initial infusion in different combinations to elucidate their effects on the mellow and thick taste of the PRT.

First, the taste of CDT, CDB, and TB in aqueous solution was evaluated (Figure 6). The mellow and thick taste of the TB solution was significantly stronger than that of the PRT. CDT and CDB solutions did not taste mellower or thicker than PRT. Next, the taste of CDT, CDB, and TB in the PRT was evaluated. It was found that their taste in PRT were significantly different from that in aqueous solution. TB tasted mellower and thicker in the PRT compared with that in aqueous solution at the same concentration. The sensory evaluation of the solution with different blends of CDT, CDB, and TB species was carried out, and the mellow and thick taste increased from 4.45 ± 0.17 to 4.52 ± 0.04, 5.13 ± 0.39, and 4.90 ± 0.24, respectively. The taste of combinations of two of the three candidates in the PRT was also evaluated. Among these combinations, Combination 5 (CDT + TB) had the strongest mellow and thick taste, and Combination 6 (CDB + TB) ranked second. Finally, the taste of the PRT with the addition of all three candidates was evaluated. Interestingly, the mellow and thick scores were not significantly changed before or after the addition.

The above results show that TB contributed to the mellow and thick taste. CDT and CDB individually attenuated the contribution of TB to the mellow and thick taste. CDT and CDB together totally counteracted the mellow and thick taste caused by TB. The results suggest that the taste of compounds was affected by the food system and that interactions among components were complicated. Further studies, such as how CDT and CDB affect the mellow and thick taste of TB, whether different polysaccharides and proteins have similar effects, and whether the ratio of CDT and CDB plays a role, will be conducted in the future.

## 4. Conclusions

Based on the content and distribution of chemical components of SGT and PRT, combined with sensory evaluation, the findings can be summarized as follows.

It was found that the pile fermentation process caused a series of complex and intense biotransformation reactions and oxidation reactions in SGT, which led to the reduction of its PP, SP, SS, and catechin content, as well as the production of a large amount of GA and TB.

The mellow and thick taste of the PRT, RL, and FL had a great difference. From the foregoing discussion, the macromolecular compounds without molecular interception of 10 kDa might play a key role in the mellow and thick taste of PRT. Further taste recombination experiments showed that TB, CDB, and CDT improved the mellow and thick taste to varying degrees. The improvement effect of PRT with the addition of TB was the best, increasing the score from 4.45 to 5.13. These results can provide useful information for taste improvement and taste research of PRT products. They also provide a solid theoretical basis for the processing of PRT.

## Figures and Tables

**Figure 1 foods-11-02285-f001:**
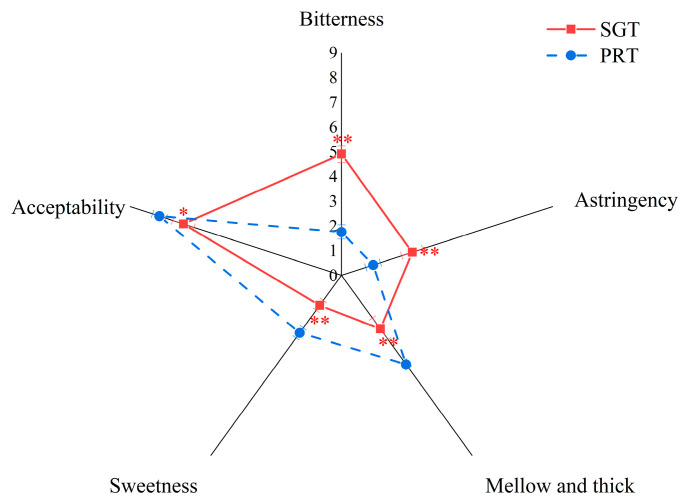
Sensory evaluation of SGT and PRT initial infusions. SGT: sun−dried green tea; PRT: pu−erh ripe tea; ** in the same taste attribute indicates *p* < 0.01; * indicates *p* < 0.05.

**Figure 2 foods-11-02285-f002:**
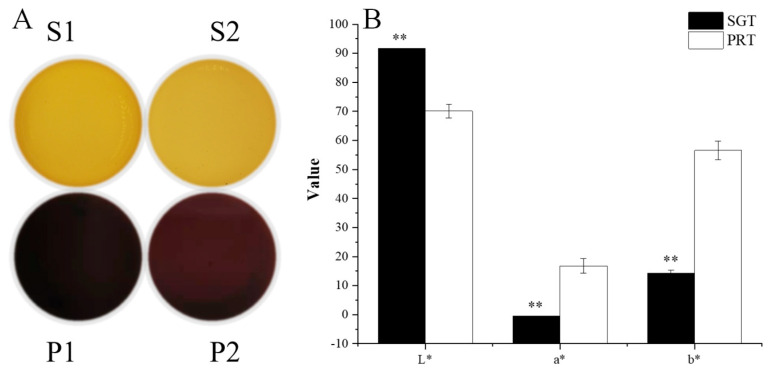
The color of SGT and PRT initial infusions. (**A**) Visual appearance. (**B**) Chromatic parameters; ** indicates *p* < 0.01. SGT: sun−dried green tea; PRT: pu−erh ripe tea.

**Figure 3 foods-11-02285-f003:**
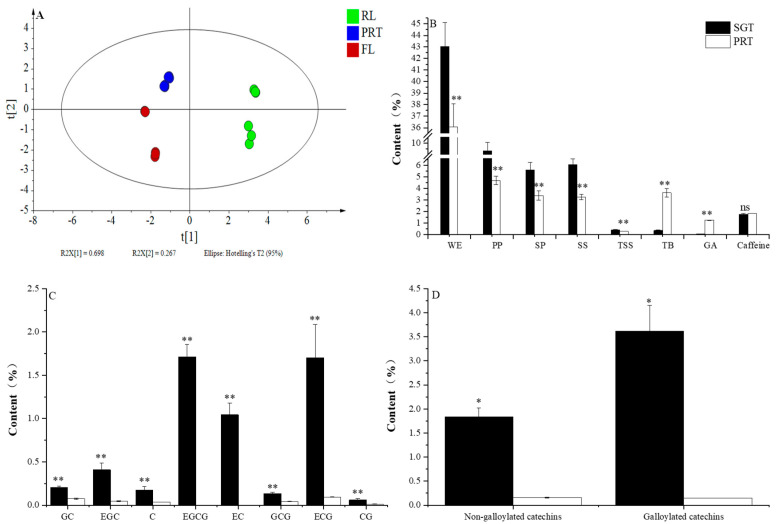
Analysis of physicochemical composition. (**A**) Analysis results of PLS−DA. The score scatter plots of SGT and PRT. (**B**) PP: total tea polyphenols; SP: soluble proteins; SS: total soluble sugars; TSS: total soluble solids; TB: theabrownin; GA: gallic acid; and caffeine. (**C**) Catechins. (**D**) Non−galloylated catechins and galloylated catechins. SGT: sun−dried green tea; PRT: pu−erh ripe tea; ** indicates *p* < 0.01; * indicates *p* < 0.05; ns indicates *p* > 0.05.

**Figure 4 foods-11-02285-f004:**
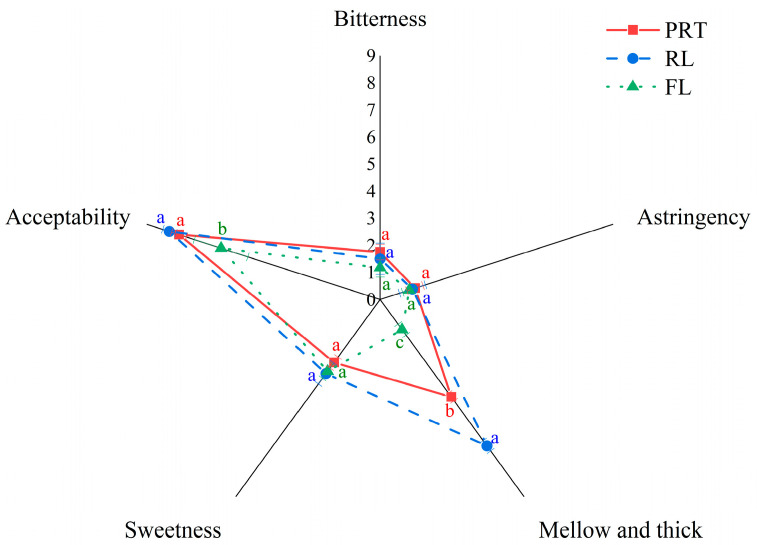
Contributions of PRT, RL, and FL to the mellow and thick taste. PRT: pu−erh ripe tea; RL: rejection liquid, FL: filtration liquid. Different lowercase letters in the same taste attribute indicate significant differences between data (*p* < 0.05).

**Figure 5 foods-11-02285-f005:**
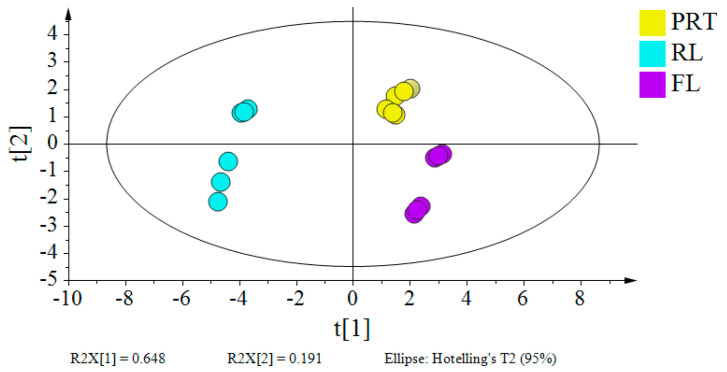
A plot of partial−least−squares discriminant analysis (PLS−DA) based on the chemical composition of PRT initial infusion, RL, and FL. PRT: pu−erh ripe tea; RL: rejection liquid; FL: filtration liquid.

**Figure 6 foods-11-02285-f006:**
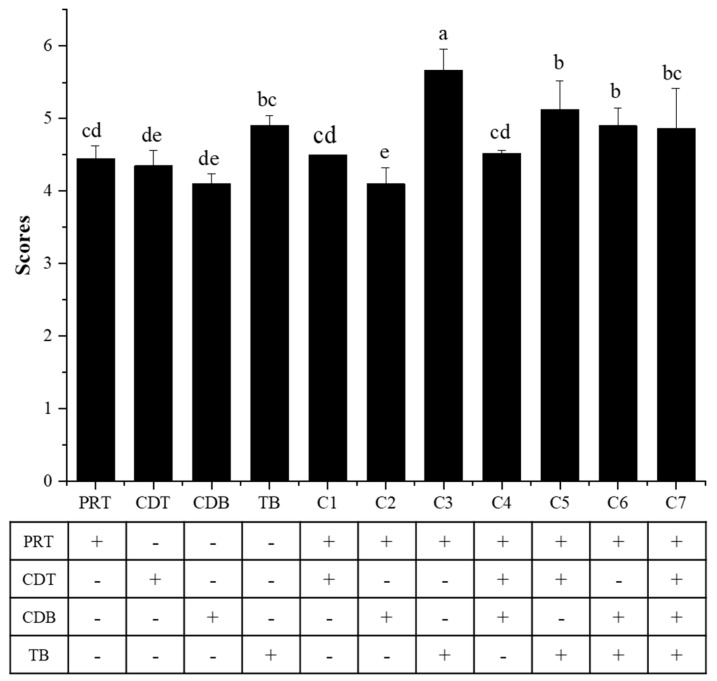
The mellow and thick taste evaluations of PRT initial infusions and recombination. Different lowercase letters indicate significant differences between data (*p* < 0.05). PRT: pu−erh ripe tea; CDT: crude polysaccharides; CDB: crude protein; TB: theabrownin; Cn: composite n. + indicates addition, − indicates no addition.

**Table 1 foods-11-02285-t001:** Information on the tea samples.

Sample Type	Sample Number	Information
SGT (Sun−dried green tea)	S1	SGT (Rong’s No. 1 Pile in 2020)
	S2	SGT (Rong’s No. 2 Pile in 2020)
PRT (Pu−erh ripe tea)	P1	PRT (Fermentation of Rong’s No. 1 Pile in 2020)
	P2	PRT (Fermentation of Rong’s No. 2 Pile in 2020)

**Table 2 foods-11-02285-t002:** Experimental treatment with different formulations.

Number	Treatment	Formulations
1	PRT (P1, CK)	10 mg·mL^−1^
2	CDB	1 mg·mL^−1^
3	CDT	1 mg·mL^−1^
4	TB	1 mg·mL^−1^
5	Composite 1	PRT + 1 mg·L^−1^ CDT
6	Composite 2	PRT + 1 mg·L^−1^ CDB
7	Composite 3	PRT + 1 mg·L^−1^ TB
8	Composite 4	PRT + 1 mg·L^−1^ CDT + 1 mg·L^−1^ CDB
9	Composite 5	PRT + 1 mg·L^−1^ CDT + 1 mg·L^−1^ TB
10	Composite 6	PRT + 1 mg·L^−1^ CDB + 1 mg·L^−1^ TB
11	Composite 7	PRT + 1 mg·L^−1^ CDT + 1 mg·L^−1^ CDB + 1 mg·L^−1^ TB

**Table 3 foods-11-02285-t003:** Distribution of the physicochemical components of the PRT initial infusion, RL, and FL.

Physicochemical Component (mg·L^−1^)	PRT Initial Infusion		RL		FL	
	Concentration	Distribution Ratio (PRT/PRT, %)	Concentration	Distribution Ratio (RL/PRT, %)	Concentration	Distribution Ratio (FL/PRT, %)
Total tea polyphenols	416.20 ± 34.47 ^b^	100.00	1032.78 ± 159.26 ^a^	247.02	154.77 ± 7.39 ^c^	37.45
Soluble protein	300.15 ± 37.12 ^b^	100.00	733.43 ± 74.34 ^a^	245.84	73.38 ± 25.85 ^c^	24.06
Total soluble sugar	288.55 ± 21.43 ^b^	100.00	1727.56 ± 194.87 ^a^	597.88	176.03 ± 29.59 ^b^	60.81
Total soluble solids (%)	0.29 ± 0.00 ^b^	100.00	1.01 ± 0.08 ^a^	352.73	0.19 ± 0.01 ^c^	66.02
TB (%)	3.63 ± 0.37 ^b^	100.00	5.62 ± 0.04 ^a^	154.82	1.12 ± 0.06 ^c^	30.85
Gallic acid	110.79 ± 2.52 ^a^	100.00	87.95 ± 9.65 ^c^	79.29	102.14 ± 9.46 ^b^	92.38
Caffeine	162.82 ± 0.45 ^a^	100.00	161.53 ± 0.22 ^b^	99.21	146.02 ± 1.06 ^c^	89.68
GC	6.68 ± 0.80 ^a^	100.00	5.43 ± 0.87 ^b^	81.15	6.29 ± 0.28 ^b^	94.82
EGC	4.14 ± 0.50 ^a^	100.00	0.81 ± 0.48 ^b^	19.26	3.25 ± 1.15 ^a^	80.24
C	3.14 ± 0.07 ^a^	100.00	3.14 ± 0.08 ^a^	100.19	3.12 ± 0.05 ^a^	99.43
EGCG	−	−	−	−	−	−
EC	−	−	−	−	−	−
GCG	3.82 ± 0.12 ^b^	100.00	4.10 ± 0.51 ^a^	107.04	3.79 ± 0.12 ^c^	99.12
ECG	8.36 ± 0.28 ^b^	100.00	11.57 ± 1.14 ^a^	138.25	3.45 ± 0.39 ^c^	41.49
CG	1.09 ± 0.23 ^b^	100.00	1.63 ± 0.22 ^a^	156.10	0.39 ± 0.11 ^c^	36.03
Non−galloylated catechins	13.96 ± 0.82 ^a^	100.00	9.38 ± 1.35 ^b^	66.99	12.65 ± 1.02 ^a^	91.17
Galloylated catechins	13.27 ± 0.17 ^b^	100.00	17.29 ± 0.77 ^a^	130.34	7.64 ± 0.61 ^c^	57.56

“−”, Information was not found in the literature. Different lowercase letters in the same row indicate significant differences between data (*p* < 0.05). PRT: pu−erh ripe tea; RL: rejection liquid, FL: filtration liquid.

## Data Availability

The data presented in this study are available on request from the corresponding author.

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
