# Peer review of "Mellow and Thick Taste of Pu−Erh Ripe Tea Based on Chemical Properties by Sensory−Directed Flavor Analysis"

_foods, 2022, doi:10.3390/foods11152285_

Round 1
Reviewer 1 Report
Line 62: the sentence itself feels like it is missing something. Either remove the such as or finish the thought by adding what other macro polymers can be found in the tea.
I would suggest having someone read over the manuscript for grammatical errors and missing words.
Line 65 - 66: I would suggest moving rearranging relationship the status, to read the relationship status.
Line 72: Instead of if, use "of"
Line 125: Use author's last name and not their first.
Line 127: I believe it was a typo but you need to remove the extra period.
Reviewer 2 Report
The aim of the work described in this manuscript is the study of the composition of pu-erh ripe tea in order to provide knowledge about the biochemical determinants of the peculiar “mellow” and “thick” taste.
The work is interesting, but in my opinion this manuscript is very hard to read, since I had some difficult to follow a train of thought. There are repeated paragraphs.
I think that more attention should be payed to the measurement and the description of biochemical composition data. Moreover, the volatile fraction, which is an important driver for sensory characteristics, should be taken into consideration.
Lines 49-51: please clarify. Did you indicated a complex among polysaccharides, phenols and peptides?
Line 63: such as?
Line 99: was “overall taste” evaluated in terms of intensity? or hedonistic? Since in line 153 a higher value is described as “better”. Why did you judged useful this descriptor?
Section 2.3: please add to this section the number of replicates.
Line 113: Robert’s method was developed for the study of theaflavins and thearubigins. Did you adapt the method for theabrownins? Is the theabrownins absorption spectrum similar or different to that of the other compounds?
Line 143: please change “the analysis of significant difference” to “the analysis of variance”, and add the used test.
Results: I suggest to organize this section in a different way, since you state that the work is focused on chemical composition and on its effects on taste. I suggest to begin from chemical composition data of each product.
Figure 1 and 4: please indicate the statistic significance among samples for the scores of each descriptor.
Lines 307-316 are repeated (the same as 247-256) The reference 27 is not the mentioned study.
Lines 261-262: please, clarify. Was the concentration of TB inferred? since the great importance of this component, I think it is better to perform the concentration analysis.
Round 2
Reviewer 2 Report
The Authors have carefully considered all remarks and the quality of the manuscript is surely improved. Moreover, all issues have been addressed. In my opinion, this interesting manuscript is suitable for publication.